# Normal Variance Mixture with Arcsine Law of an Interpolating Walk Between Persistent Random Walk and Quantum Walk

**DOI:** 10.3390/e27070670

**Published:** 2025-06-23

**Authors:** Saori Yoshino, Honoka Shiratori, Tomoki Yamagami, Ryoichi Horisaki, Etsuo Segawa

**Affiliations:** 1Graduate School of Environment and Information Sciences, Yokohama National University, Hodogaya, Yokohama 240-8501, Japan; 2Department of Information Physics and Computing, The University of Tokyo, Bunkyo, Tokyo 113-8656, Japan

**Keywords:** limit theorem, persistent (or correlated) random walk, discrete-time stochastic quantum walk

## Abstract

We propose a model that interpolates between quantum walks and persistent (correlated) random walks using one parameter on the one-dimensional lattice. We show that the limit distribution is described by the normal variance mixture with the arcsine law.

## 1. Introduction

Discrete-time quantum walk on the one-dimensional lattice with the uniform quantum coin is one of the first triggers that led to the development of studies on discrete-time quantum walks, e.g., [1,2,3,4,5]. The time evolution is determined by the 2-dimensional unitary matrix$$C=\left[\begin{array}{cc} a & b\\ c & d \end{array}\right].$$Note that the unitarity of *C* implies ${\left|a\right|}^{2}={\left|d\right|}^{2}$, ${\left|b\right|}^{2}={\left|c\right|}^{2}$ and ${\left|a\right|}^{2}+{\left|b\right|}^{2}={\left|c\right|}^{2}+{\left|d\right|}^{2}=1$ and so on. The diagonal elements of *C* represent the complex-valued weights associated with the transmission in the same direction (left or right) as that of the previous step, while the off-diagonal elements represent the complex-valued weights associated with the reflection to the different direction from that of the previous step at each vertex. Let us denote the time evolution of this quantum walk by $L^{QW}$. On the other hand, we choose the corresponding random walk model as persistent random walk (or equivalently correlated random walk) [6,7], where the probabilities of moving left and right are determined by the choice of the directions at the previous time. In this paper, we set that the moving probability at each time step and each position of the correlate random walk is locally represented by$$\begin{array}{c} C\circ \bar{C}=\left[\begin{array}{cc} {\left|a\right|}^{2} & {\left|b\right|}^{2}\\ {\left|c\right|}^{2} & {\left|d\right|}^{2} \end{array}\right]. \end{array}$$The total time evolution operator of such a persistent random walk mode is denoted by $L^{RW}$. Since both dynamics evolve on the *arcs* of the one-dimensional lattice, we find that it is possible to represent $L^{QW}$ and $L^{RW}$ so that both domains are the same and both operators preserve the trace. (See Section 3 for more detail.)

So we propose the following natural model which interpolates between the persistent random walk and the quantum walk. At each time step independently, a walker flips a Bernoulli coin for the choice of $L^{RW}$ or $L^{QW}$; that is, a walker chooses $L^{RW}$ with probability *p*, while it chooses $L^{QW}$ with probability q=1−p. Let $\mu_{t}^{\left(p\right)}$ be the average of the distributions of the walker’s position at time *t* with respect to all the possible coin tosses and $X_{t}^{\left(p\right)}$ of a position of an interpolate walker at time *t*, that is, $X_{t}^{\left(p\right)}∼\mu_{t}^{\left(p\right)}$. Here, if a random variable *X* follows a distribution *F*, we write X∼F in short. The distribution $\mu_{t}^{\left(p\right)}$ can be realized by taking the trace of the *t*-th iteration of the convex combination of $L^{RW}$ and $L^{QW}$:$$L_{p}:=pL^{RW}+qL^{QW}.$$We emphasize that this model can be regarded as a discrete-time model analogous to the Lindblad master equation for the *quantum stochastic walk* [8,9] on a one-dimensional lattice. See [8] for more detail, which is discussed in the general connected graphs. Under the difference equation of the quantum stochastic walk given by [8], Equation (32), a discrete-time quantum stochastic walk driven by a convex combination of quantum walks and correlated random walks can be constructed on general graphs. Note that in this case, the graph G in [8] should not be converted to G itself but to $\vec{L}\left(G\right)$, where $\vec{L}\left(G\right)=\left(\vec{X},\vec{A}\right)$ is the *directed* line graph of G; that is, $\vec{X}=A$, and $\left(a,b\right)\in \vec{A}$ iff the terminal of *a* coincides with the origin of *b* in G. Thus, following the framework presented in [8], one can also extend our model on Z to higher dimensions $Z^{d}\;\left(d\geq 2\right)$. We study such a discrete-time stochastic quantum walk model focusing on a simple graph Z to see its fundamental property.

In this paper, we characterize the limit behavior of the quantum stochastic walk model on the one-dimensional lattice by the normal variance mixture with the arcsine law as follows.

**Theorem** **1**(Main theorem)**.** *Let ν be the scaled arcsine distribution, such that*$$\nu \left(du\right)=\frac{1_{\left(\frac{1-q}{1+q}r,\frac{1+q}{1-q}r\right)}\left(u\right)}{\pi \sqrt{\left|u-\frac{1-q}{1+q}r\right|\left|u-\frac{1+q}{1-q}r\right|}}du,$$*where $r={\left|a\right|}^{2}/{\left|b\right|}^{2}$, and $1_{A}\left(u\right)$ is the indicator function of the set A⊂R. Let $\Sigma_{\nu }^{2}$ be a random variable following ν. A position of an interpolating walker with the parameter p∈(0,1) at time t is denoted by $X_{t}^{\left(p\right)}$ with the initial state $\rho_{0}\left(x\right)=\left(1/2\right)\delta_{0}\left(x\right)I_{2}$. Then, $X_{t}^{\left(p\right)}/\sqrt{t}$ converges in distribution to the average of $N(0,\Sigma_{\nu }^{2})$ in ν, that is,*
$$\frac{X_{t}^{\left(p\right)}}{\sqrt{t}}\;∼\;E_{\nu }\left(\;N\left(0,\Sigma_{\nu }^{2}\right)\;\right)\;\;\left(t\to \infty \right),$$*where $E_{\nu }\left(\cdot \right)$ is the average in ν.*

The following corollaries are equivalent expressions to Theorem 1.

**Corollary** **1.**
*Let $Y_{*}^{\left(p\right)}$ be a random variable following the limit distribution of $X_{t}^{\left(p\right)}/\sqrt{t}$(t→∞). The limit density function of $Y_{*}^{\left(p\right)}$ is expressed by*

(1)
$$f_{*}\left(x\right)=E_{\nu }\left[\frac{e^{-\frac{x^{2}}{2\Sigma_{\nu }^{2}}}}{\sqrt{2\pi \Sigma_{\nu }^{2}}}\right]=\int_{-\infty }^{\infty }\frac{e^{-\frac{x^{2}}{2u}}}{\sqrt{2\pi u}}\;\frac{1_{\left(\frac{1-q}{1+q}r,\frac{1+q}{1-q}r\right)}\left(u\right)}{\pi \sqrt{\left|u-\frac{1-q}{1+q}r\right|\left|u-\frac{1+q}{1-q}r\right|}}du.$$



**Corollary** **2.**
*Let Z be a random variable following the normal distribution N(0,1). Assume that the random variables $\sigma^{\left(p\right)}:=\left|\sqrt{\Sigma_{\nu }^{2}}\right|$ and Z are independent. Then the (cumulative) distribution of $Y_{*}^{\left(p\right)}$ is described by*

$$P\left(Y_{*}^{\left(p\right)}\leq x\right)=P\left(\sigma^{\left(p\right)}Z\leq x\right),$$

*for any x∈R, which is nothing but the mixing variance of the normal distribution [10].*


The variance of the limit distribution is given by a random variable following the arcsine distribution, where the highest probability is taken around both end points, that is, r(1−q)/(1+q) and r(1+q)/(1−q). When *q* approaches 0, ν can be regarded as a delta function concentrated at ${\left|b\right|}^{2}/{\left|a\right|}^{2}$, which is consistent with the limiting distribution of the (purely) persistent random walk [7]. On the other hand, as *q* approaches 1, the domain of the density function that gives the variance spans all positive real numbers, reflecting the linear spreading of the quantum walk at q=1.

Such a crossover between the quantum walk and the persistent (or correlated) random walk models is discussed in [11] by a geometric control, and it is shown that a glimpse of the ballistic spreading of the quantum walk is already seen even in the geometry where the walk is close to the random walk. On the other hand, there are other types of quantum walk models that exhibit essentially diffusive spreading, for example, [12,13]. In [12], the quantum walk with decoherence is proposed as well as the diffusive spreading of the Hadamard-based walk. It seems that this model can be reproduced by setting |a|=1/2 in our walk model. Note that when |a|=1/2, the persistent random walk is simply reduced to the isotropic random walk. We connect the persistent random walk, which is a *double* Markov process, and the quantum walk by considering 0<|a|<1 and characterize the interpolating walk model by the weak convergence. In [13], the random choice of unitary operators at each time step induces decoherence and, as a result, the diffusive spreading exhibits a drift. Recently, another discrete-time quantum stochastic walk model on a 1-dimensional lattice was proposed, and the recurrence properties were discussed in [14].

This paper is organized as follows. In Section 2, we give a numerical demonstration of the consistency of our main results of Theorem 1. In Section 3, we give a detailed definition of our interpolating walk model. In Section 3, we show the proofs of the main results. Section 4 presents the summary and discussion. The main theorem gives the discontinuity of the limit theorem with respect to the parameter *p* at p=0. Then we discuss the limit theorem for *p* close to 0.

## 2. Numerical Demonstrations

Let us see the demonstrations of the interpolating walks with the parameters *p* given by the numerical simulations and the consistency of Theorem 1.

First, we show the distributions $\mu_{t}^{\left(p\right)}$ of the interpolating walks given by the numerical simulations with $a=b=c=-d=1/\sqrt{2}$ and the parameters p=1/100, 1/10, 1/2 and 9/10 in Figure 1a–d, respectively, at the final time s=100. We can observe a transition from distributions of the quantum walk to the random walk as *p* increases. The shape of the distribution rapidly gets closer to a normal distribution along with the growth of the value of *p*; the peak around the origin is already higher than those around the edges in p=1/10 shown in Figure 1b. This observation supports our statement of the main theorem that once the rate of decoherence *p* deviates from 0 slightly, the shape of the limit distribution of $X_{t}^{\left(p\right)}/\sqrt{t}$ has the structure of a normal distribution.

Secondly, let us discuss the consistency of our analytical results with Theorem 1 in more detail. Note that if x+t is odd, then $\mu_{t}\left(x\right)=0$ for any t∈N and x∈Z. The following step function $f_{t}$, which satisfies $\int_{-\infty }^{\infty }f_{t}\left(x\right)dx=1$ and reflects such a parity with the position and time, should approximate the limit density function $f_{*}$ of $E_{\nu }\left(\;N\left(0,\Sigma_{\nu }^{2}\right)\;\right)$ in Theorem 1 in that(2)$$\lim_{t\to \infty }P\left(X_{t}/\sqrt{t}<x\right)=\lim_{t\to \infty }\int_{-\infty }^{x}f_{t}\left(u\right)du=\int_{-\infty }^{x}f_{*}\left(u\right)du,$$
for any x∈R:$$f_{t}\left(u\right)=\frac{\sqrt{t}}{2}\times \left\{\begin{array}{cc} \mu_{t}\left(-t\right) & :\;-\sqrt{t}-\frac{1}{\sqrt{t}}<u\leq -\sqrt{t}+\frac{1}{\sqrt{t}},\\ \mu_{t}\left(-t+2\right) & :\;-\sqrt{t}+\frac{1}{\sqrt{t}}<u\leq -\sqrt{t}+\frac{3}{\sqrt{t}},\\ \;\vdots \\ \mu_{t}\left(t-2\right) & :\;\sqrt{t}-\frac{3}{\sqrt{t}}<u\leq \sqrt{t}-\frac{1}{\sqrt{t}},\\ \mu_{t}\left(t\right) & :\;\sqrt{t}-\frac{1}{\sqrt{t}}<u\leq \sqrt{t}+\frac{1}{\sqrt{t}},\\ 0 & :\;\mathrm{otherwise}. \end{array}\right.$$Figure 2 shows the simultaneous plot of $f_{t}$ and $f_{*}$ for t=100 and p=1/2. Here we use the second equality in Equation (Equation 1) to approximately draw the limit density function of $Y_{*}^{\left(p\right)}=\sigma^{\left(p\right)}Z$, $f_{*}$, by the Gauss–Kronrod quadrature approximation [15]. We see that the step function $f_{t}\left(x\right)$ appears to be close to $f_{*}\left(x\right)$ for sufficiently large *t*, as expected with Equation (Equation 2).

Finally, let us discuss the case that the parameter *p* is very close to 0. It is well-known that if the parameter p=0, $X_{s}^{\left(p\right)}/s$ converges to the Konno function [16]. On the other hand, if the parameter *p* is even slightly greater than 0, the shape of the limit distribution of $X_{s}^{\left(p\right)}/\sqrt{s}$ has the structure of a normal distribution by our main theorem. In this sense, Theorem 1 indicates that the limit distribution in p>0 is clearly distinguished from that in p=0. However, in the above setting, it is assumed that the parameter *p* is independent of the final time *s* even if *p* is small, which implies that the decoherence occurs infinitely many times when s→∞. Then we set the parameter *p* by λ/s with some positive constant value λ>0 to avoid the infinitely many uses of the evolution operator $L^{RW}$. Note that such a setting gives the Poisson occurrence of $L^{RW}$ because the probability that “the number of choices of the evolution operator $L^{RW}$ during the time span *s* is *k*” can be expressed byskpk(1−p)s−k→e−λλkk!(s→∞).Then the number of occurrence $L^{RW}$ during the infinite time span is finite (its average is λ), and the time interval at which the evolution $L^{RW}$ is used follows the exponential distribution with parameter λ. We discuss the behavior of the interpolating walk in such a natural setting of p=λ/s in Section 5 and obtain an expression of the characteristic function of $X_{t}^{\left(p\right)}/t$ in the long-time limit, which shows the ballistic spreading in Proposition 3. Figure 3 shows a distribution $\mu_{s}^{\left(1/s\right)}$ at the final time *s* with the parameter p=1/s. A “protrusion” is seen in the middle of a typical quantum walk distribution. Its characterization is left as an open problem in this paper.

## 3. Setting of Interpolating Walk

### 3.1. Random Walk

Let $A_{Z}$ be the set of symmetric arcs of Z. Each element of $A_{Z}$ is denoted by (x,x−1)(x∈Z). Here the origin and terminus of (x,x−1) are *x* and x−1, respectively. Note that the inverse of (x,x−1) is (x−1,x). The total state space is set by $\ell^{2}\left(A_{Z}\right)$. For each time step t=0,1,2,… with the initial state $\psi_{0}\in \ell^{2}\left(A_{Z}\right)$, $\psi_{t}\in \ell^{2}\left(A_{Z}\right)$ is obtained by the following iterations: $\psi_{t+1}=M\psi_{t}$. Here the local time evolution at x∈Z is denoted by(3)$$\psi_{t+1}\left(x,x-1\right)=r\psi_{t}\left(x+1,x\right)+\left(1-r\right)\psi_{t}\left(x-1,x\right)$$(4)$$\psi_{t+1}\left(x,x+1\right)=\left(1-r\right)\psi_{t}\left(x+1,x\right)+r\psi_{t}\left(x-1,x\right)$$
for every x∈Z and t∈N. It moves as shown in Figure 4 and Figure 5. This random walk model is called a persistent random/correlated random walk. At each time, the random walker chooses the same direction as the one chosen exactly 1 step before (right or left) with probability *r*, and a different direction with probability 1−r. The probability of being found at vertex *x* at time *t* is defined as(5)$$\mu_{t}^{RW}\left(x\right)=\psi_{t}\left(x,x-1\right)+\psi_{t}\left(x,x+1\right).$$Let $M_{2}\left(C\right)$ and U(2) be the sets of the 2-dimensional complex-valued matrices and the unitary matrices. Let us consider an equivalent expression to this random walk on the space ${\left(M_{2}\left(C\right)\right)}^{Z\times Z}$ as follows. Set 2×2 matrices P and Q as$$P=\left[\begin{array}{cc} a & b\\ 0 & 0 \end{array}\right],\;Q=\left[\begin{array}{cc} 0 & 0\\ c & d \end{array}\right],$$
where $P+Q=\left[\begin{array}{cc} a & b\\ c & d \end{array}\right]\in U\left(2\right)$ with ${\left|a\right|}^{2}={\left|d\right|}^{2}=r$, ${\left|b\right|}^{2}={\left|c\right|}^{2}=1-r$. Define $L^{RW}:{\left(M_{2}\left(C\right)\right)}^{Z\times Z}\to {\left(M_{2}\left(C\right)\right)}^{Z\times Z}$ by(6)$$\left(L^{RW}\rho \right)\left(x,y\right)=P\rho \left(x+1,y+1\right)P^{*}+Q\rho \left(x-1,y-1\right)Q^{*}$$
for any $\rho \in {\left(M_{2}\left(C\right)\right)}^{Z\times Z}$ and (x,y)∈Z×Z. For each time step t=0,1,2,… with the initial state $\rho_{0}\in {\left(M_{2}\left(C\right)\right)}^{Z\times Z}$, $\rho_{t}\in {\left(M_{2}\left(C\right)\right)}^{Z\times Z}$ is obtained by $\rho_{t+1}=L^{RW}\rho_{t}$. Thus this walk with the time evolution $L^{RW}$ can be regarded as an open quantum random walk [17]. Let us see that this open quantum random walk is isomorphic to the persistent random walk as follows. The time iteration can be reexpressed by$$\begin{array}{cc} \rho_{t+1}\left(x,x\right) & =P\rho_{t}\left(x+1,x+1\right)P^{*}+Q\rho_{t}\left(x-1,x-1\right)Q^{*}\\ \; & =\langle -|\rho_{t}\left(x+1,x+1\right)|-\rangle |L\rangle \langle L|+\langle +|\rho_{t}\left(x-1,x-1\right)|+\rangle |R\rangle \langle R|. \end{array}$$The second equality derives from P=|L〉〈−|,Q=|R〉〈+|, where we set |L〉,|R〉,|−〉,|+〉 as$$|L\rangle =\left[\begin{array}{c} 1\\ 0 \end{array}\right],\;|R\rangle =\left[\begin{array}{c} 0\\ 1 \end{array}\right],\;|-\rangle =\left[\begin{array}{c} \bar{a}\\ \bar{b} \end{array}\right],\;|+\rangle =\left[\begin{array}{c} \bar{c}\\ \bar{d} \end{array}\right].$$Therefore, the following equation is derived:$$\langle -|\rho_{t+1}\left(x,x\right)|-\rangle =r\langle -|\rho_{t}\left(x+1,x+1\right)|-\rangle +\left(1-r\right)\langle +|\rho_{t}\left(x-1,x-1\right)|+\rangle$$$$\langle +|\rho_{t+1}\left(x,x\right)|+\rangle =r\langle -|\rho_{t}\left(x+1,x+1\right)|-\rangle +\left(1-r\right)\langle +|\rho_{t}\left(x-1,x-1\right)|+\rangle ,$$
which is the same as the local time evolution at x∈Z on the space $\ell^{2}\left(A_{Z}\right)$ (Equation 3) and (Equation 4). Indeed we obtain the following proposition.

**Proposition** **1.**
*For a persistent random walk with the initial state $\mu_{t}^{RW}\left(0\right)=\delta_{0}\left(x\right){\left[\begin{array}{cc} \alpha & \beta \end{array}\right]}^{⊤}$, if $\rho_{0}\left(x,y\right)=\delta_{0,0}\left(x,y\right)\rho_{0}$ satisfies $\langle -|\rho_{0}|-\rangle =\alpha$ and $\langle +|\rho_{0}|+\rangle =\beta$, then for all t=0,1,2,…,*

$$\mu_{t}^{RW}\left(x\right)=\mathrm{tr}\left[\rho_{t}\left(x,x\right)\right].$$

*Here,*

$$|-\rangle ={\left[\begin{array}{cc} a & b \end{array}\right]}^{*},\;|+\rangle ={\left[\begin{array}{cc} c & d \end{array}\right]}^{*}.$$



**Proof.** By calculating the trace of ρ in this case, we obtain$$\begin{array}{cc} \mathrm{tr}[\;\rho_{t}\left(x,x\right)] & =\mathrm{tr}[\rho_{t}\left(x,x\right)\left(\;|-\rangle \langle -|+|+\rangle \langle +|\;\right)\;]\\ \; & =\mathrm{tr}\left[\rho_{t}\left(x,x\right)|-\rangle \langle -|\right]+tr\left[\rho_{t}\left(x\right)|+\rangle \langle +|\right]\\ \; & =\mathrm{tr}\left[\langle -|\rho_{t}\left(x,x\right)|-\rangle \right]+tr\left[\langle +|\rho_{t}\left(x,x\right)|+\rangle \right]\\ \; & =\langle -|\rho_{t}\left(x,x\right)|-\rangle +\langle +|\rho_{t}\left(x,x\right)|+\rangle , \end{array}$$
so if we set $\rho_{0}$ such that it satisfies$$\left[\begin{array}{c} \alpha \\ \beta \end{array}\right]=\left[\begin{array}{c} \langle -|\rho_{0}\left(x,x\right)|-\rangle \\ \langle +|\rho_{0}\left(x,x\right)|+\rangle \end{array}\right]$$
when $\left[\begin{array}{c} \alpha \\ \beta \end{array}\right]$ is given as the initial state, the finding probability $\mu_{t}^{RW}\left(x\right)$ defined in (Equation 5) for each time step t=0,1,2,… is given by$$\mu_{t}^{RW}\left(x\right)=\mathrm{tr}\left[\rho_{t}\left(x,x\right)\right].$$□

### 3.2. Quantum Walk

As with the random walk discussed in the previous section, first, the total state space is set by $\ell^{2}\left(A_{Z}\right)$. For each time step t=0,1,2,… with the initial state $\psi_{0}\in \ell^{2}\left(A_{Z}\right)$, $\psi_{t}\in \ell^{2}\left(A_{Z}\right)$ is obtained by the following iterations: $\psi_{t+1}=U\psi_{t}$. Here the local time evolution at x∈Z is denoted by(7)$$\psi_{t+1}\left(x+1,x\right)=a\psi_{t}\left(x+2,x+1\right)+b\psi_{t}\left(x,x+1\right)$$(8)$$\psi_{t+1}\left(x-1,x\right)=c\psi_{t}\left(x,x-1\right)+d\psi_{t}\left(x-2,x-1\right)$$
for every x∈Z and t∈N. It moves as shown in Figure 6 and Figure 7. Here a,b,c,d∈C and $C=\left[\begin{array}{cc} a & b\\ c & d \end{array}\right]$ is unitary. The left-moving quantum walker chooses to move left with probability amplitude *a* and right with probability amplitude *c*, while the right-moving quantum walker chooses to move left with probability amplitude *b* and right with probability amplitude *d*. The probability of being found at vertex *x* at time *t* is defined as(9)$$\mu_{t}^{QW}\left(x\right)=|\psi_{t}{\left(x+1,x\right)|}^{2}+{\left|\psi_{t}\left(x-1,x\right)\right|}^{2}.$$Let us consider an equivalent expression to this quantum walk on the space ${\left(M_{2}\left(C\right)\right)}^{Z\times Z}$ as follows. Define $L^{QW}:{\left(M_{2}\left(C\right)\right)}^{Z\times Z}\to {\left(M_{2}\left(C\right)\right)}^{Z\times Z}$ by(10)$$\begin{array}{c} \left(L^{QW}\rho \right)\left(x,y\right)=P\rho \left(x+1,y+1\right)P^{*}+Q\rho \left(x-1,y-1\right)Q^{*}\\ \;+P\rho \left(x+1,y-1\right)Q^{*}+Q\rho \left(x-1,y+1\right)P^{*} \end{array}$$
for any $\rho \in {\left(M_{2}\left(C\right)\right)}^{Z\times Z}$. For each time step t=0,1,2,… with the initial state $\rho_{0}\in {\left(M_{2}\left(C\right)\right)}^{Z\times Z}$, $\rho_{t}\in {\left(M_{2}\left(C\right)\right)}^{Z\times Z}$ is given by the recursion $\rho_{t+1}=L^{QW}\rho_{t}$. In particular, if we set $\rho_{t}^{\prime }\left(x,y\right):=|\psi_{t}\left(x\right)\rangle \langle \psi_{t}\left(y\right)|$, it satisfies $L^{QW}\rho_{t}^{\prime }=\rho_{t+1}^{\prime }$ because$$\begin{array}{cc} \; & |\psi_{t+1}\left(x\right)\rangle \langle \psi_{t+1}\left(x\right)|\\ \; & =(\;P|\psi_{t}\left(x+1\right)\rangle +Q|\psi_{t}\left(x-1\right)\rangle \;)\;(\;P|\psi_{t}\left(y+1\right)\rangle +Q|\psi_{t}\left(y-1\right){\rangle \;)}^{*}\\ \; & =P|\psi_{t}\left(x+1\right)\rangle \langle \psi_{t}\left(y+1\right)|P^{*}+Q|\psi_{t}\left(x-1\right)\rangle \langle \psi_{t}\left(y-1\right)|Q^{*}\\ \; & \;\;\;+P|\psi_{t}\left(x+1\right)\rangle \langle \psi_{t}\left(y-1\right)|Q^{*}+Q|\psi_{t}\left(x-1\right)\rangle \langle \psi_{t}\left(y+1\right)|P^{*}. \end{array}$$This corresponds to the local time evolution at x∈Z on the space $\ell^{2}\left(A_{Z}\right)$ (Equation 7) and (Equation 8).

**Proposition** **2.**
*For a quantum walk with the initial state $\mu_{t}^{QW}\left(0\right)=\delta_{0}\left(x\right){\left[\begin{array}{cc} \alpha & \beta \end{array}\right]}^{⊤}$, if $\rho_{0}\left(x,y\right)=\delta_{0,0}\left(x,y\right)\rho_{0}$ satisfies $\rho_{0}=|0\rangle \langle 0|⊗{\left[\begin{array}{cc} \alpha & \beta \end{array}\right]}^{⊤}\left[\begin{array}{cc} \bar{\alpha } & \bar{\beta } \end{array}\right]$, then for all t=0,1,2,…,*

$$\mu_{t}^{QW}\left(x\right)=\mathrm{tr}\left[\rho_{t}\left(x,x\right)\right].$$



**Proof.** By the induction with respect to *t*, we have $\rho_{t}\left(x,y\right)=|\psi_{t}\left(x\right)\rangle \langle \psi_{t}\left(y\right)|$ for any t≥0. Then taking the trace of $\rho_{t}\left(x,x\right)$, we obtain$$\begin{array}{cc} \mathrm{tr}[\rho_{t}\left(x,x\right)] & =\mathrm{tr}[|\psi_{t}\left(x\right)\rangle \langle \psi_{t}\left(x\right)|]=\mathrm{tr}[\langle \psi_{t}\left(x\right),\psi_{t}\left(x\right)\rangle ]\\ \; & =\left|\right|\psi_{t}\left(x\right){\left|\right|}^{2}\\ \; & =\mu_{t}^{QW}\left(x\right), \end{array}$$
which implies the desired conclusion. □

### 3.3. Interpolating Walk

We define the interpolation model between the random walk and the quantum walk as follows. For a parameter 0≤p≤1, let $L_{p}$ be defined as$$L_{p}:=\left(1-p\right)L^{QW}+pL^{RW}.$$Set(11)$$\rho_{t}:=L_{p}\rho_{t-1},$$
with the initial state $\rho_{0}\left(x,y\right)=\left(1/2\right)\delta_{\left(0,0\right)}\left(x,y\right)I_{2}$. Since $L_{p}$ is a convex combination of $L^{QW}$ in (Equation 10) and $L^{RW}$ in (Equation 6) which are trace-preserving, then at each time t=0,1,2,…, the finding probability $\mu_{t}\left(x\right)$ at each position *x* can be defined as(12)$$\mu_{t}\left(x\right)=\mathrm{tr}\left[\rho_{t}\left(x,x\right)\right].$$The operator $L_{p}$ is expressed by$$\begin{array}{c} L_{p}\rho \left(x,y\right)=\left\{P\rho \left(x+1,y+1\right)P^{*}+Q\rho \left(x-1,y-1\right)Q^{*}\right\}\\ \;+q\left\{P\rho \left(x+1,y-1\right)Q^{*}+Q\rho \left(x-1,y+1\right)P^{*}\right\}. \end{array}$$Then the first and second terms are interpreted as the decoherence and interference terms, respectively, and the strength of the interference is tunable by the parameter q=1−p∈[0,1].

The following observation is well-known, but let us confirm that this equation represents a process in which a random walk is performed with probability *p*, while a quantum walk is executed with probability q=1−p. Here, we set i.i.d. Bernoulli sequence ω=(ω(1),ω(2),…,ω(n)) such that ω(t)∈{RW,QW}(t=1,2,…,n) and$$\rho_{n}^{\omega }:=L^{\omega (n-1)}\rho_{n-1}^{\omega }$$
with $\rho_{0}^{\omega }=\rho_{0}$. By taking the average over $\omega \in {\left\{RW,QW\right\}}^{n}$, we have$$E\left[\rho_{n}^{\omega }\right]=E\left[L^{\omega (n-1)}\rho_{n-1}^{\omega }\right]=\left(\left(1-p\right)L^{QW}+pL^{RW}\right)\;E\left[\rho_{n-1}^{\omega }\right].$$Since this equation satisfies the same recurrence relation as (Equation 11), the interpolation model can be interpreted as a walk that represents the average behavior of a system that randomly chooses the evolution of the random walk with probability *p* or that of the quantum walk with probability 1−p at each time step independently.

## 4. Proof of Theorem 1

### 4.1. Fourier Transform

The following isomorphism is useful for our analysis: for any matrices A,B and ρ of size *N*,$$\begin{array}{cc} {\left(A\rho B\right)}_{l,m} & =\sum_{i,j}{\left(A\right)}_{l,i}{\left(\rho \right)}_{i,j}{\left(B\right)}_{j,m}\\ \; & =\sum_{i,j}{\left(A\right)}_{l,i}{\left(B^{⊤}\right)}_{m,j}{\left(\rho \right)}_{i,j}\\ \; & =\sum_{i,j}{\left(A⊗B^{⊤}\right)}_{\left(l,m),(i,j\right)}{\left(\rho \right)}_{i,j}, \end{array}$$
where $B^{⊤}$ is the matrix transpose of *B*. Therefore when the matrix $\rho \in M_{N}\left(C\right)$ is reinterpreted as a vector $\rho \in C^{N\times N}$ as ${\left(\rho \right)}_{i,j}=\vec{\rho }\left(i,j\right)$, we have$$A\rho B≅\left(A⊗B^{⊤}\right)\vec{\rho }.$$From now on, denote the vector representation of $\rho \left(x,y\right)\in M_{2}\left(C\right)$ by $\vec{\rho }\left(x,y\right)\in C^{4}$.

Let $X_{t}$ be a random variable following the distribution $\mu_{t}$. To obtain the limit distribution of $X_{t}$, we concentrate on obtaining the asymptotics of the characteristic function of $X_{t}$,$$E\left[e^{i\xi X_{t}}\right]=\sum_{x\in Z}\mu_{t}\left(x\right)e^{i\xi x}.$$The following lemma plays an important role in computing the characteristic function.

**Lemma** **1.**
*Set ${\vec{\rho }}_{0}\left(k,l\right)=\sum_{x,y\in Z}{\vec{\rho }}_{0}\left(x,y\right)e^{ikx}e^{ily}$ for any k,l∈[0,2π) and*

$${\hat{H}}_{p}\left(k,l\right)=e^{-i(k+l)}P⊗\bar{P}+e^{i(k+l)}Q⊗\bar{Q}+\left(1-p\right)\left(e^{-i(k-l)}P⊗\bar{Q}+e^{i(k-l)}Q⊗\bar{P}\right).$$

*Then, we have*

$$E\left[e^{i\xi X_{t}}\right]=\left(\langle LL\right|+\langle RR|)\int_{0}^{2\pi }{\hat{H}}_{p}^{t}\left(\xi -k,k\right){\vec{\rho }}_{0}\left(\xi -k,k\right)\frac{dk}{2\pi }.$$



**Proof.** The finding probability $\mu_{t}\left(x\right)=\mathrm{tr}\left[\rho_{t}\left(x,x\right)\right]$ defined in (Equation 12), so the characteristic function takes the following form:$$E\left[e^{i\xi X_{t}}\right]=\sum_{x}\mathrm{tr}\left[\rho_{t}\left(x,x\right)\right]e^{i\xi x}=\mathrm{tr}\left[\sum_{x}\rho_{t}\left(x,x\right)e^{i\xi x}\right].$$Set the Fourier transform $\left(F\rho_{t}\right)\left(k,l\right):=\sum_{x,y}\rho_{t}\left(x,y\right)e^{ikx}e^{ily}=:{\hat{\rho }}_{t}\left(k,l\right)$ (k,l∈[0,2π)). Then the following equation is obtained:$$\begin{array}{cc} \int_{0}^{2\pi }{\hat{\rho }}_{t}\left(\xi -k,k\right)\frac{dk}{2\pi } & =\int_{0}^{2\pi }\sum_{x,y}\rho_{t}\left(x,y\right)e^{i(\xi -k)x}e^{iky}\frac{dk}{2\pi }\\ & =\sum_{x,y}\rho_{t}\left(x,y\right)e^{i\xi x}\int_{0}^{2\pi }e^{ik\xi (y-x)}\frac{dk}{2\pi }\\ & =\sum_{x}\rho_{t}\left(x,x\right)e^{i\xi x}. \end{array}$$This implies that the characteristic function of $\rho_{t}$ can be expressed by(13)$$E\left[e^{i\xi X_{t}}\right]=\mathrm{tr}\int_{0}^{2\pi }{\hat{\rho }}_{t}\left(\xi -k,k\right)\frac{dk}{2\pi }.$$By the recurrence relation representing the local time evolution,$$\rho_{t+1}\left(x,y\right)=L_{p}\rho_{t}\left(x,y\right)=\left(1-p\right)L^{QW}\rho_{t}\left(x,y\right)+pL^{RW}\rho_{t}\left(x,y\right),$$${\hat{\rho }}_{t}\left(k,l\right)$ satisfies the following recursion with respect to the time step *t*:$$\begin{array}{cc} {\hat{\rho }}_{t+1}\left(k,l\right) & =\left(e^{-ik}P\right){\hat{\rho }}_{t}\left(k,l\right)\left(e^{-il}P^{*}\right)+\left(e^{ik}Q\right){\hat{\rho }}_{t}\left(k,l\right)\left(e^{il}Q^{*}\right)\\ \; & +\left(1-p\right)\{\left(e^{-ik}P\right){\hat{\rho }}_{t}\left(k,l\right)\left(e^{il}Q^{*}\right)+\left(e^{ik}Q\right){\hat{\rho }}_{t}\left(k,l\right)\left(e^{-il}P^{*}\right)\}, \end{array}$$
which implies(14)$$\begin{array}{c} {\vec{\rho }}_{t}\left(k,l\right)={\hat{H}}_{p}\left(k,l\right){\vec{\rho }}_{t-1}\left(k,l\right)={\hat{H}}_{p}^{t}\left(k,l\right){\vec{\rho }}_{0}\left(k,l\right). \end{array}$$Inserting (Equation 14) into (Equation 13), we obtain the desired conclusion. □

### 4.2. Asymptotics of the Eigenvalues of ${\hat{H}}_{p}\left(\xi /\sqrt{t}-k,k\right)$

To obtain the limit of the characteristic function of $X_{t}/\sqrt{t}$, we consider the expansion of the time evolution operator in the Fourier space ${\hat{H}}_{p}\left(\xi /\sqrt{t}-k,k\right)$ in Lemma 1 as follows so that the Kato perturbation theorem [18] can be applied.$$\begin{array}{cc} {\hat{H}}_{p}\left(\xi /\sqrt{t}-k,k\right) & =e^{-i(\xi /\sqrt{t})}P⊗\bar{P}+e^{i(\xi /\sqrt{t})}Q⊗\bar{Q}\\ \; & \;+\left(1-p\right)\left(e^{-i(\xi /\sqrt{t}-2k)}P⊗\bar{Q}+e^{i(\xi /\sqrt{t}-2k)}Q⊗\bar{P}\right)\\ \; & =T+\frac{1}{\sqrt{t}}T^{\left(1\right)}+\frac{1}{t}T^{\left(2\right)}+O\left(\frac{1}{{\left(\sqrt{t}\right)}^{3}}\right), \end{array}$$where(15)$$T=P⊗\bar{P}+Q⊗\bar{Q}+\left(1-p\right)\left(e^{2ik}P⊗\bar{Q}+e^{-2ik}Q⊗\bar{P}\right),$$(16)$$T^{\left(1\right)}=i\xi \left\{-P⊗\bar{P}+Q⊗\bar{Q}+\left(1-p\right)\left(-e^{2ik}P⊗\bar{Q}+e^{-2ik}Q⊗\bar{P}\right)\right\}=i\xi DT,$$(17)$$T^{\left(2\right)}=-\frac{1}{2}\xi^{2}\left\{P⊗\bar{P}+Q⊗\bar{Q}+\left(1-p\right)\left(e^{2ik}P⊗\bar{Q}+e^{-2ik}Q⊗\bar{P}\right)\right\}=-\frac{1}{2}\xi^{2}T.$$Here, *D* is defined as $D=\left[\begin{array}{cc} -1 & 0\\ 0 & 1 \end{array}\right]⊗I$.

Then let us put $T\left(\epsilon \right):={\hat{H}}_{p}\left(\epsilon \xi -k,k\right)$. We are interested in the expansion of the perturbed eigenvalues λ(ϵ) in spec(T(ϵ)) which split from the non-perturbed eigenvalue λ:=λ(0) by the small perturbation ϵ. To this end, we check the spectral formation of the non-perturbed matrix *T*.

**Lemma** **2.**
*Let T be the non-perturbed matrix of T(ϵ). Then 1∈spec(T) and*

$$\dim\left(\ker\left(1-T\right)\right)=\left\{\begin{array}{cc} 1 & :\;p\neq 0,\;\\ 2 & :\;p=0.\; \end{array}\right.$$

*The vector ${\left[1\;0\;0\;1\right]}^{⊤}\in C^{4}$ is an eigenvector of T, which is independent of the parameter p. Moreover, for any λ∈spec(T)∖{1},*

$$\left|\lambda \right|\left\{\begin{array}{cc} <1 & :\;p\neq 0,\;\\ =1 & :\;p=0.\; \end{array}\right.$$



**Proof.** The non-perturbed matrix *T* is reexpressed by$$T=\left(I-p\Pi_{LR,RL}\right)\left(D\left(k\right)⊗D\left(-k\right)\right)\left(C⊗\bar{C}\right),$$
where $\Pi_{LR,RL}$ is the projection onto span{|LR〉,|RL〉} and$$D\left(k\right)=\left[\begin{array}{cc} e^{-ik} & 0\\ 0 & e^{ik} \end{array}\right].$$Note that $\left||(I-p\Pi_{LR,RL})\psi ||\leq ||\psi |\right|$ for any $\psi \in C^{4}$ and the equality holds if and only if p=0 or $\psi \in \mathrm{Ran}(I-\Pi_{LR,RL})$. By the properties of the unitarity of *C*, for example, ${\left|a\right|}^{2}+{\left|b\right|}^{2}={\left|c\right|}^{2}+{\left|d\right|}^{2}=1$ and $a\bar{b}+c\bar{d}=0$, only the eigenvector of *T* which belongs to $\mathrm{Ran}(I-\Pi_{LR,RL})$ is ${\left[1\;0\;0\;1\right]}^{⊤}$. This implies that if p≠0, then dimker(1−T)=1. On the other hand, if p=0, then *T* becomes $\left(D\left(k\right)C\right)⊗\bar{\left(D(k)C\right)}$. Since D(k)C is a unitary, all the eigenvalues of *T* are unitary, in particular, dimker(1−T)=2. This finishes the proof. □

Let us concentrate on the p≠0 case, since the p=0 case reproduces the usual quantum walk driven by the unitary operator. By Lemma 2, in a large time iteration of T(ϵ), the eigenspace of the eigenvalue λ(ϵ), which splits from the non-perturbed eigenvalue 1 of *T* by the small perturbation ϵ, contributes almost all the behavior. The following lemma gives the expansion of λ(ϵ) with respect to ϵ.

**Lemma** **3.**
*Assume p>0. Let λ(ϵ) be the eigenvalue of T(ϵ) which splits from the non-perturbed eigenvalue 1 of T. Then we have*

$$\begin{array}{c} \lambda \left(\epsilon \right)=1-\epsilon^{2}\;\left(\frac{r}{2}\;\frac{1+q^{2}-2q\cos2l}{1-q^{2}}\xi^{2}\right)+O\left(\epsilon^{3}\right) \end{array}$$



**Proof.** The eigenvalues of ${\hat{H}}_{p}$ that are close to 1 can be expanded as follows because 1 is a simple eigenvalue of *T*:(18)$$\lambda \left(\epsilon \right)=1+\epsilon \lambda^{\left(1\right)}+\epsilon^{2}\lambda^{\left(2\right)}+O\left(\epsilon^{3}\right).$$Here, since λ=1 is simple, according to [18], $\lambda^{\left(1\right)}$ and $\lambda^{\left(2\right)}$ coincide with the coefficients of the second and third orders of the weighted mean of the eigenvalues that form the λ-group, (λ=1);$$\lambda^{\left(1\right)}=\mathrm{tr}\left[T^{\left(1\right)}\Pi_{0}\right]\;\mathrm{and}\;\lambda^{\left(2\right)}=\mathrm{tr}\left[T^{\left(2\right)}\Pi_{0}-T^{\left(1\right)}ST^{\left(1\right)}\Pi_{0}\right],$$
respectively, where $\Pi_{0}$ is the eigenprojection associated with the eigenvalue 1 of *T*, and *S* is the reduced resolvent associated with the eigenvalue 1 of *T*. By using (Equation 15) and (Equation 16), then we have $\Pi_{0}=1/2{\left[\begin{array}{cccc} 1 & 0 & 0 & 1 \end{array}\right]}^{⊤}\left[\begin{array}{cccc} 1 & 0 & 0 & 1 \end{array}\right]$, and(19)$$\begin{array}{c} \lambda^{\left(1\right)}=\mathrm{tr}\left[T^{\left(1\right)}\Pi_{0}\right]=\mathrm{tr}\left[i\xi DT\Pi_{0}\right]=\mathrm{tr}\left[i\xi D\Pi_{0}\right]=\mathrm{tr}\left[\frac{i\xi }{2}\left[\begin{array}{cc} -1 & 0\\ 0 & 1 \end{array}\right]⊗I\right]=0. \end{array}$$Let *S* be defined as $S=-\sum_{i=1}^{3}\Pi_{i}/\left(1-\lambda_{i}\right)$, where $\lambda_{i}$ are the eigenvalues of *T* that are not equal to 1, and $P_{i}$ are the corresponding eigenprojections. $\lambda^{\left(2\right)}$ is expressed as follows by using (Equation 15)–(Equation 17):(20)$$\begin{array}{cc} \lambda^{\left(2\right)}\; & =\mathrm{tr}[T^{\left(2\right)}\Pi_{0}-T^{\left(1\right)}ST^{\left(1\right)}\Pi_{0}]\\ \; & =\mathrm{tr}\left[-\frac{1}{2}\xi^{2}T\Pi_{0}-\xi^{2}DT\sum_{i=1}^{3}\frac{\Pi_{i}}{1-\lambda_{i}}DT\Pi_{0}\right]\\ \; & =\mathrm{tr}\left[-\frac{1}{2}\xi^{2}T\Pi_{0}\right]-\mathrm{tr}\left[\xi^{2}DT\sum_{i=1}^{3}\frac{\Pi_{i}}{1-\lambda_{i}}DT\Pi_{0}\right]. \end{array}$$Since $T\Pi_{i}=\lambda_{i}\Pi_{i}$ (i=0,1,2,3), the first and second terms of (Equation 20) are reduced to(21)$$\begin{array}{cc} \mathrm{the}\;\mathrm{first}\;\mathrm{term}:\;\; & \mathrm{tr}\left[-\frac{1}{2}\xi^{2}T\Pi_{0}\right]=\mathrm{tr}\left[-\frac{1}{2}\xi^{2}\Pi_{0}\right]=-\frac{1}{2}\xi^{2}, \end{array}$$(22)$$\begin{array}{cc} \mathrm{the}\;\mathrm{second}\;\mathrm{term}:\;\; & \mathrm{tr}\left[\xi^{2}DT\sum_{i=1}^{3}\frac{\Pi_{i}}{1-\lambda_{i}}DT\Pi_{0}\right]=\mathrm{tr}\left[\xi^{2}\sum_{i=1}^{3}\frac{\lambda_{i}}{1-\lambda_{i}}D\Pi_{i}D\Pi_{0}\right]. \end{array}$$Let us continue the computation of the second term as follows. Due to the commutativity of the trace operation, it is shown that$$\begin{array}{cc} \mathrm{tr}\left[\xi^{2}\sum_{i=1}^{3}\frac{\lambda_{i}}{1-\lambda_{i}}D\Pi_{i}D\Pi_{0}\right] & =\mathrm{tr}\left[\xi^{2}\sum_{i=1}^{3}\frac{\lambda_{i}}{1-\lambda_{i}}\Pi_{i}D\Pi_{0}D\right]\\ & =\mathrm{tr}\left[\frac{\xi^{2}}{2}\sum_{i=1}^{3}\frac{\lambda_{i}}{1-\lambda_{i}}\Pi_{i}\left[\begin{array}{c} -1\\ 0\\ 0\\ 1 \end{array}\right]\left[\begin{array}{cccc} -1 & 0 & 0 & 1 \end{array}\right]\right]. \end{array}$$Note that by setting $T^{\prime }=T-\Pi_{0}$, we have$$\sum_{i=1}^{3}\frac{\lambda_{i}}{1-\lambda_{i}}\Pi_{i}={\left(1-T^{\prime }\right)}^{-1}T^{\prime }.$$Then let us compute ${\left(1-T^{\prime }\right)}^{-1}T^{\prime }$ as follows. In this case, with q=1−p, *T* is expressed as $T=P⊗\bar{P}+Q⊗\bar{Q}+q\left(e^{2ik}P⊗\bar{Q}+e^{-2ik}Q⊗\bar{P}\right)$, and thus we have$$T^{\prime }=T-\Pi_{0}=\left[\begin{array}{cccc} {\left|a\right|}^{2}-\frac{1}{2} & a\bar{b} & \bar{a}b & {\left|b\right|}^{2}-\frac{1}{2}\\ -e^{2il}q\bar{a}b & e^{2il}q{\left|a\right|}^{2} & -e^{2i(l-\sigma )}qb^{2} & e^{2il}q\bar{a}b\\ -e^{-2il}qa\bar{b} & -e^{-2i(l-\sigma )}q{\bar{b}}^{2} & e^{-2il}q{\left|a\right|}^{2} & e^{-2il}qa\bar{b}\\ {\left|b\right|}^{2}-\frac{1}{2} & -a\bar{b} & -\bar{a}b & {\left|a\right|}^{2}-\frac{1}{2} \end{array}\right]$$
by utilizing the unitarity of the matrix *C*. Here, $e^{i\sigma },$$e^{i\delta }$, and *l* are defined as $a=\left|a\right|e^{i\sigma },$$e^{i\delta }=ad-bc,$ and l=k+σ−δ/2. By performing calculations using cofactor expansion, the following result is obtained:$$\begin{array}{cc} {\left(1-T^{\prime }\right)}_{1,1}^{-1} & ={\left(1-T^{\prime }\right)}_{4,4}^{-1}\\ \; & =\frac{1}{{2|b|}^{2}\left(1-q^{2}\right)}\\ \; & \;\;\;\;\;\;\times \left[\left(\frac{3}{2}-{\left|a\right|}^{2}\right)\left\{1+q^{2}{\left(|a\right|}^{2}-{\left|b\right|}^{2}{)-2q|a|}^{2}\cos2l\right\}+{2q|a|}^{2}{\left|b\right|}^{2}\left(\cos2l-q\right)\right],\\ {\left(1-T^{\prime }\right)}_{1,2}^{-1} & =-{\left(1-T^{\prime }\right)}_{2,1}^{-1}\\ & =\frac{1}{{2|b|}^{2}\left(1-q^{2}\right)}\left(1-e^{-2il}q\right)a\bar{b},\\ {\left(1-T^{\prime }\right)}_{1,3}^{-1} & =-{\left(1-T^{\prime }\right)}_{3,1}^{-1}\\ & =\frac{1}{{2|b|}^{2}\left(1-q^{2}\right)}\left(1-e^{2il}q\right)\bar{a}b,\\ {\left(1-T^{\prime }\right)}_{1,4}^{-1} & ={\left(1-T^{\prime }\right)}_{4,1}^{-1}\\ \; & =\frac{1}{{2|b|}^{2}\left(1-q^{2}\right)}\\ \; & \;\;\;\;\;\;\times \left[\left(\frac{1}{2}-{\left|a\right|}^{2}\right)\left\{1+q^{2}{\left(|a\right|}^{2}-{\left|b\right|}^{2}{)-2q|a|}^{2}\cos2l\right\}+{2q|a|}^{2}{\left|b\right|}^{2}\left(\cos2l-q\right)\right]. \end{array}$$Therefore, simplifying slightly, we obtain the following formula:$$\begin{array}{cc} {\left({\left(1-T^{\prime }\right)}^{-1}T^{\prime }\right)}_{1,1} & =-{\left({\left(1-T^{\prime }\right)}^{-1}T^{\prime }\right)}_{1,4}=-{\left({\left(1-T^{\prime }\right)}^{-1}T^{\prime }\right)}_{4,1}={\left({\left(1-T^{\prime }\right)}^{-1}T^{\prime }\right)}_{4,4}\\ \; & =\frac{1}{{2|b|}^{2}\left(1-q^{2}\right)}\left\{\frac{1}{2}\left(q^{2}-1\right)+{\left|a\right|}^{2}\left(1-q\cos2l\right)\right\}. \end{array}$$Put $r={\left|a\right|}^{2}/{\left|b\right|}^{2}$. The above formula simplifies the expression of the second term by(23)$$\begin{array}{cc} \mathrm{the}\;\mathrm{second}\;\mathrm{term} & =\mathrm{tr}\left[\frac{\xi^{2}}{2}{\left(1-T^{\prime }\right)}^{-1}T^{\prime }\left[\begin{array}{c} -1\\ 0\\ 0\\ 1 \end{array}\right]\left[\begin{array}{cccc} -1 & 0 & 0 & 1 \end{array}\right]\right]\\ \; & =\frac{\xi^{2}}{{2|b|}^{2}\left(1-q^{2}\right)}\left\{q^{2}-1+2{\left|a\right|}^{2}\left(1-q\cos2l\right)\right\}. \end{array}$$Inserting the first term (Equation 21) and the second term (Equation 23) into (Equation 20), we obtain(24)$$\begin{array}{cc} \lambda^{\left(2\right)} & =-\frac{1}{2}\xi^{2}-\frac{\xi^{2}}{{2|b|}^{2}\left(1-q^{2}\right)}\left\{q^{2}-1+2{\left|a\right|}^{2}\left(1-q\cos2l\right)\right\}\\ & =-\frac{r(1+q^{2}-2q\cos2l)}{2(1-q^{2})}\xi^{2}. \end{array}$$The eigenvalues of ${\hat{H}}_{p}$ that are close to 1 can be analyzed as follows by inserting (Equation 19) and (Equation 24) into (Equation 18):$$\begin{array}{c} \lambda \left(\epsilon \right)=1-\epsilon^{2}\frac{r}{2}\frac{1+q^{2}-2q\cos2l}{1-q^{2}}\xi^{2}+O\left(\epsilon^{3}\right). \end{array}$$□Now by using Lemmas 1–3, let us finish the proof of Theorem 1.

### 4.3. Proof of Theorem 1

**Proof.** Since the initial state is defined by $\rho_{0}\left(x,y\right)I_{2}$, note that the initial state in the Fourier space, ${\vec{\rho }}_{0}\left(k,l\right)$, is given by$${\vec{\rho }}_{0}\left(k,l\right)=\frac{1}{2}\left(\;|LL\rangle +|RR\rangle \;\right)$$
for any k,l∈[0,2π). Then using Lemmas 1, the characteristic function can be expressed as follows:$$\begin{array}{cc} E[e^{i\xi X_{t}/\sqrt{t}}] & =\left(\;\langle LL|+\langle RR|\;\right)\int_{0}^{2\pi }{\vec{\rho }}_{t}\left(\xi /\sqrt{t}-k,k\right)\frac{dk}{2\pi }\\ \; & =\left(\;\langle LL|+\langle RR|\;\right)\int_{0}^{2\pi }H_{p}^{t}\left(\xi /\sqrt{t}-k,k\right)\;{\vec{\rho }}_{0}\left(\xi /\sqrt{t}-k,k\right)\frac{dk}{2\pi }\\ \; & \to \int_{0}^{2\pi }e^{-\frac{1}{2}\frac{r(1+q^{2}-2q\cos2l)}{1-q^{2}}\xi^{2}}\frac{dl}{2\pi }\;\;\;\;\left(t\to \infty \right) \end{array}$$Here in the last equation, we used Lemmas 2, 3 and the fact that eigenvalues other than those in the neighborhood of 1 approach zero in the limit of t→∞ due to the Riemann–Lebesgue lemma. By Lemma 3, for the eigenvalues splitting from 1 by the perturbation ϵ, the second term proportional to ϵ vanishes, while the third term proportional to ϵ retains a value. To obtain a non-trivial convergence in the limit, we set ϵ by $\epsilon =1/\sqrt{t}$. Next, by setting $u=r\left(1+q^{2}-2q\cos2l\right)/\left(1-q^{2}\right)$ and performing substitution integral,$$\begin{array}{c} \int_{0}^{2\pi }e^{-\frac{1}{2}\frac{r(1+q^{2}-2q\cos2l)}{1-q^{2}}\xi^{2}}\frac{dl}{2\pi }=\int_{\frac{1-q}{1+q}r}^{\frac{1+q}{1-q}r}\frac{1}{\pi \sqrt{\left|u-\frac{1-q}{1+q}r\right|\left|u-\frac{1+q}{1-q}r\right|}}e^{-\frac{u\xi^{2}}{2}}du. \end{array}$$The term $e^{-\frac{s\xi^{2}}{2}}$ represents the form of the Fourier transform of a normal distribution, so the final form of the characteristic function in the limit is as follows:$$\begin{array}{c} \int_{\frac{1-q}{1+q}r}^{\frac{1+q}{1-q}r}\frac{1}{\pi \sqrt{\left|u-\frac{1-q}{1+q}r\right|\left|u-\frac{1+q}{1-q}r\right|}}e^{-\frac{u\xi^{2}}{2}}du=\int_{-\infty }^{\infty }e^{i\xi x}\int_{-\infty }^{\infty }\frac{e^{-\frac{x^{2}}{2u}}}{\sqrt{2\pi u}}\frac{1_{\left(\frac{1-q}{1+q}r,\frac{1+q}{1-q}r\right)}\left(u\right)}{\pi \sqrt{\left|u-\frac{1-q}{1+q}r\right|\left|u-\frac{1+q}{1-q}r\right|}}dudx. \end{array}$$In other words, the limit of the characteristic function represents the Fourier transform of the product of a normal distribution and an inverse scale factor. Here, *s* denotes the variance of the normal distribution, and the distribution of *s* follows the arcsine distribution. The resulting limit distribution corresponds to the mean of this arcsine distribution. This completes the proof. □

## 5. Summary and Discussion

In this study, we constructed a model that interpolates between the quantum walk and the random walk on a one-dimensional lattice using a single parameter p∈[0,1]. We found that the walk spreads diffusively with respect to the time step, and the limiting distribution converges to a normal variance mixture with the arcsine law depending on the parameter *p*.

Our limit theorem shows discontinuity with respect to the parameter *p* at p=0. Then future challenges include cases where the parameter *p* is very close to zero. For example, the problem under the setting of the parameter p=γ/s with some positive real value γ and the final time of the walk *s* may be interesting. Because such a setting produces a Poisson occurrence until the final time *s* of the decoherence for s→∞, the probability that “the number of occurrence of $L^{RW}$ during the time step *s* is *k*” can be described by $e^{-\gamma }\gamma^{k}/k!$ for s→∞. Indeed we obtain the following expression for the characteristic function of $X_{s}/s$ with s→∞.

**Proposition** **3.**
*Let s be the final time step and set p=γ/s with some positive real number γ>0. Put θ(k), α(k) and β(k) for k∈[0,2π) by*

$$\cos\theta (k)=|a|\cos k,\;\;\alpha \left(k\right)=|a|\sin k/\sin\theta (k),\;\;\beta \left(k\right)=\{1-\alpha {\left(k\right)}^{2}\}/2.$$

*Then we have*

(25)
$$\begin{array}{cc} \lim_{t\to \infty }E\left[e^{i\xi X_{s}/s}\right]\; & =\frac{1}{2}\int_{0}^{2\pi }e^{-\beta \left(k\right)\gamma +\sqrt{\beta {\left(k\right)}^{2}\gamma^{2}-\alpha {\left(k\right)}^{2}\xi^{2}}}\left(1+\frac{\sqrt{\beta {\left(k\right)}^{2}\gamma^{2}-\alpha {\left(k\right)}^{2}\xi^{2}}}{\beta (k)\gamma }\right)\frac{dk}{2\pi }\\ \; & \;+\frac{1}{2}\int_{0}^{2\pi }e^{-\beta \left(k\right)\gamma -\sqrt{\beta {\left(k\right)}^{2}\gamma^{2}-\alpha {\left(k\right)}^{2}\xi^{2}}}\left(1-\frac{\sqrt{\beta {\left(k\right)}^{2}\gamma^{2}-\alpha {\left(k\right)}^{2}\xi^{2}}}{\beta (k)\gamma }\right)\frac{dk}{2\pi }. \end{array}$$



**Proof.** The proof is given in Appendix A. □

**Remark** **1.***If we put γ=0 in the integral of the LHS in (Equation 25), which means p=0 (quantum walk), then*$$\begin{array}{cc} \lim_{s\to \infty }E\left[e^{i\xi X_{s}/s}\right] & =\frac{1}{2}\left(\int_{0}^{2\pi }e^{i\xi \alpha (k)}\frac{dk}{2\pi }+\int_{0}^{2\pi }e^{-i\xi \alpha (k)}\frac{dk}{2\pi }\right)\\ \; & =\int_{-\infty }^{\infty }e^{i\xi u}\frac{1_{\left(-|a|,|a|\right)}\left(u\right)}{\pi \left(1-s^{2}\right)\sqrt{{\left|a\right|}^{2}-u^{2}}}du. \end{array}$$*This is consistent with the limit theorem for the pure quantum walk [16]. On the other hand, if γ→∞, which may correspond to p=c(>0), then*$$\lim_{s\to \infty }E\left[e^{i\xi X_{s}/s}\right]=1,$$*which means that the limit density function of $X_{s}^{\left(p\right)}/s$ is $\delta_{0}\left(x\right)$. This is consistent with the “*diffusive*” spreading for the interpolating walk with a constant p in Theorem 1.*

Although the expression of Proposition 3 looks suggestive, we could not find a great interpretation of the expression. Thus it remains an open problem. The limit behavior of the interpolating walk with the parameter $p=1/s^{\alpha }$, 0<α<1 is also a future problem.

The introduction of stochasticity may influence not only the limit distribution but also the recurrence behavior of the walk in a non-trivial way. Our limit theorem implies that the scaling order of our walk is *discontinuous* with respect to the parameter *p* at p=0. On the other hand, according to the results of [14], the recurrence probability of their model with a parameter θ∈[0,π/2] is *continuous* over p∈[0,1]. Note that the model in [14] intersects with our model at θ=π/4. We expect a monotone increase in the recurrence probabilities with *p*. In contrast with our intuition, a local minimum value exists at p∈(0,1) in the recurrence probability with an appropriate parameter θ. The continuous quantum stochastic walk on the line [19] shows that the scaling order of the moment is also discontinuous with respect to p∈[0,1], but interestingly, the discontinuity appears not at p=0 but at p=1, which contrasts with our discontinuous place in the scaling of the limit theorem. More detailed analytical investigations of these intriguing behaviors are left for future work.

Additionally, while we constructed an interpolation model between persistent random walks and quantum walks on a one-dimensional lattice in this study, considering interpolation models on more general graphs under the construction method proposed by [8] and analyzing their limiting distributions also remain tasks for future research. We hope that in the near future, our observation can contribute the construction and analysis of a discrete-time version of the quantum maze problem which is explored in the continuous-time stochastic quantum walk model in [20]. 

## Figures and Tables

**Figure 1 entropy-27-00670-f001:**
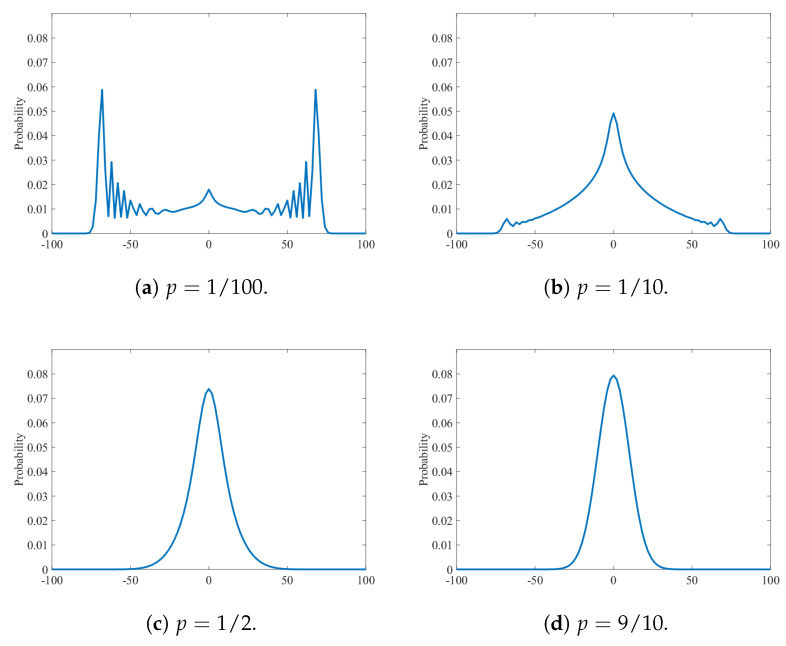
Probability distribution $\mu_{t}^{\left(p\right)}$ of the interpolating walks with $a=b=c=-d=1/\sqrt{2}$ at the final time s=100. The parameters *p* of the four panels are set to (**a**) 1/100, (**b**) 1/10, (**c**) 1/2, and (**d**) 9/10, respectively. Note that the values only on the positions labeled with even numbers are plotted; the ones on the odd-numbered positions are zero.

**Figure 2 entropy-27-00670-f002:**
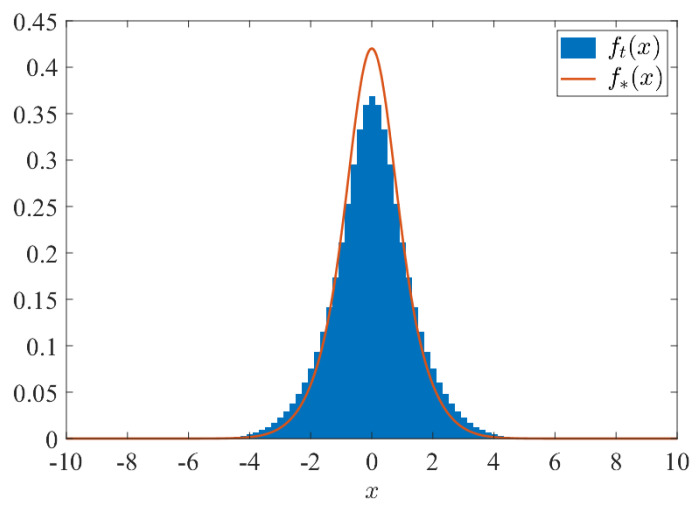
Comparison between $f_{t}\left(x\right)$ and $f_{*}\left(x\right)$: We set t=100, p=1/2 and $a=b=c=-d=1/\sqrt{2}$. The blue step function shows $f_{t}\left(x\right)$ with t=100, while the orange curve shows the limit density function $f_{*}\left(x\right)$.

**Figure 3 entropy-27-00670-f003:**
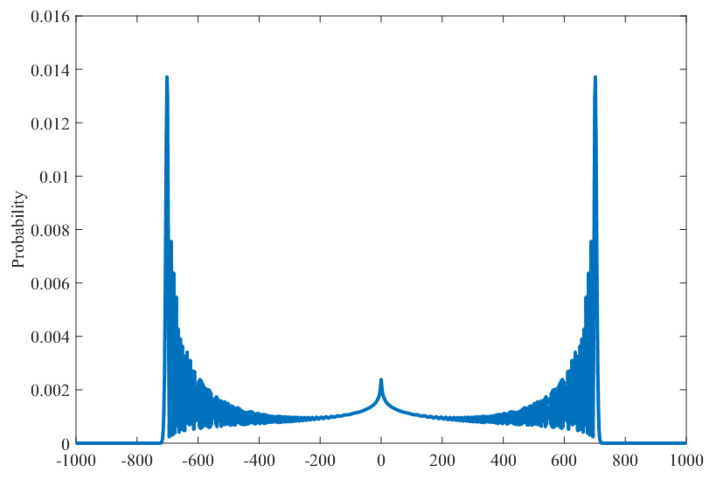
The numerical simulation of the distribution $\mu_{s}^{\left(\lambda /s\right)}$ with the parameter p=λ/s at the final time s≫1. We set λ=1, $s=10^{3}$ and $a=b=c=-d=1/\sqrt{2}$.

**Figure 4 entropy-27-00670-f004:**
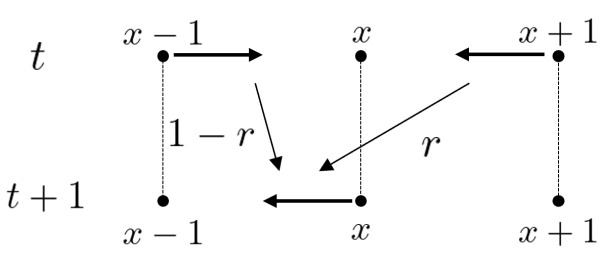
The recursion (Equation 3).

**Figure 5 entropy-27-00670-f005:**
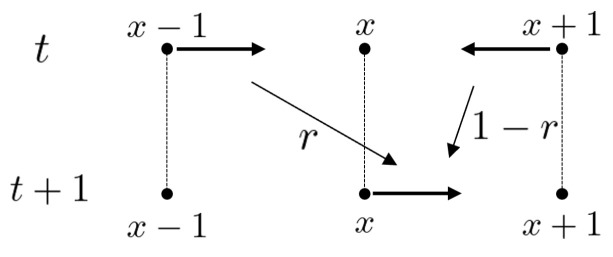
The recursion (Equation 4).

**Figure 6 entropy-27-00670-f006:**
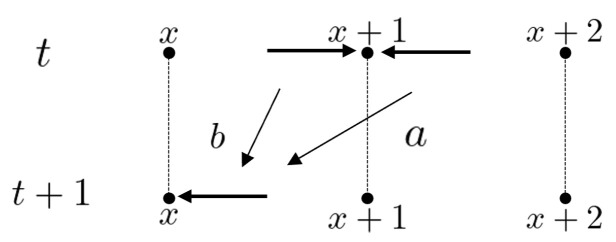
The recursion (Equation 7).

**Figure 7 entropy-27-00670-f007:**
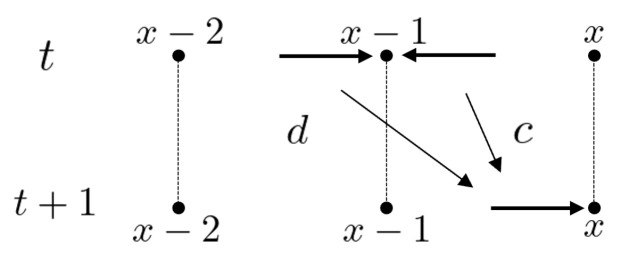
The recursion (Equation 8).

## Data Availability

The simulation results presented in this study are contained within the article. No additional data are available.

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
