# Peer review of "Normal Variance Mixture with Arcsine Law of an Interpolating Walk Between Persistent Random Walk and Quantum Walk"

_entropy, 2025, doi:10.3390/e27070670_

Round 1
Reviewer 1 Report
Comments and Suggestions for Authors In this manuscript, the authors comprehensively analyze the limiting probability distribution of a quantum walker when stochasticity is introduced into the quantum walk. Specifically, at each step, the walker is allowed, with some probability, to choose between performing a quantum walk or a correlated random walk where the direction of movement depends on the previous step. The authors demonstrate that the resulting limiting distribution is a normal variance mixture with the arcsine law. They also show that introducing a degree of stochasticity to the quantum walk leads to diffusive behavior.Although I was not able to verify all the proofs in detail, the theorems and results appear to be sound and well-supported. The manuscript is well-structured, easy to follow, and begins with a clear introduction to the topic.
I have two suggestions. First, I recommend including a brief one- or two-sentence discussion on the significance of the results and their potential applications in the context of quantum walks. For example, as the authors have already cited, introducing stochasticity may lead to interesting effects on the recurrence properties of the walk. Second, I suggest adding a short remark on whether their results are related to those obtained in the context of (continuous-time) quantum stochastic walks.
Considering the above points, I recommend this manuscript for publication in Entropy.
Author Response
Comments 1: First, I recommend including a brief one- or two-sentence discussion on the significance of the results and their potential applications in the context of quantum walks.
Response 1:Thank you for pointing this out. We agree with this comment. Therefore we inserted the following sentences in the last section on page 15 line 18 from the bottom(revised version).
The introduction of stochasticity may influence not only the limit distribution but also the
recurrence behavior of the walk in a nontrivial way. Our limit theorem implies that the scaling order
of our walk is discontinuity with respect to the parameter p at p = 0. On the other hand, according
to the results of [14], the recurrence probability of their model with a parameter θ ∈ [0, π/2] is
continuous over p ∈ [0, 1]. Note that the model in [14] intersects with our model at θ = π/4. We
expect a monotonically increasing of the recurrence probabilities with p. In contrast with such our
intuition, a local minimum value exists at p ∈ (0, 1) in the recurrence probability with an appropriate
parameter θ.
Comments 2: Second, I suggest adding a short remark on whether their results are related to those obtained in the context of (continuous-time) quantum stochastic walks.
Response 2:Thank you for pointing this out. We agree with this comment. Therefore we inserted the following sentences in the last section on page 15 line 11 from the bottom (revised version).
On the continuous quantum stochastic walk on the line [19] shows that the scaling order
of the moment is also discontinuous with respect to p ∈ [0, 1], but interestingly, the discontinuous
appears at not p = 0 but p = 1, which contrasts with our discontinuous place in the scaling of the
limit theorem. A more detailed analytical investigations of these intriguing behaviors are left for
future work.
Reviewer 2 Report
Comments and Suggestions for Authors
The paper is well written and merits publication in its current form.
Author Response
Comment 1: The paper is well written and merits publication in its current form.
Response 1: Thank you for your review.
Reviewer 3 Report
Comments and Suggestions for Authors
In this manuscript, the authors proposed a model that interpolates between the quantum walk and the random walk on the one-dimensional lattice, and presented a thorough analysis of the model. This model can build a link between quantum random walk and classical random walk, and would be helpful for understanding quantum walk models and developing quantum walk-based algorithms. Before I can recommend its publication on Entropy, there are questions to be addressed by the authors first.
- Quantum stochastic walk is also a model including quantum random walk and classical random walk. What are the differences between QSW and the model here, and what are the relation between them? It should be thoroughly analyzed.
- How is the model extended to higher dimensions, like 2D or 3D lattices? This would affect the potentials in applications of the model here.
- The possible physical implementations for the model could be discussed.
- Some typos are found including equations print in line 167, line 132, etc.
Author Response
Comments 1: Quantum stochastic walk is also a model including quantum random walk and classical random walk. What are the differences between QSW and the model here, and what are the relation between them? It should be thoroughly analyzed.
Response 1:Thank you for pointing this out. We agree with this comment. Therefore we inserted the following sentences in the first section on page 2 line 15 (revised version).
[8] for more detail,
which is discussed in general connected graphs. Under the difference equation of the quantum
stochastic walk given by [8, Eq. (32)], a discrete-time quantum stochastic walk driven by a convex
combination of quantum walks and correlated random walks can be constructed on general graphs.
Note that in this case the graph G in [8] should be converted to not G itself but L(G), where
L(G) = (X, A) is the directed line graph of G, that is, X = A, and (a, b) ∈ A iff the terminal of
a coincides with the origin of b in G.
Comments 2: How is the model extended to higher dimensions, like 2D or 3D lattices? This would affect the potentials in applications of the model here.
Response 2:Thank you for pointing this out. We agree with this comment. Therefore we inserted the following sentences in the first section on page 2 line 21 (revised version).
Thus following the framework presented in [8], one can also
extend our model on Z to higher dimensions Zd (d ≥ 2). We study such a discrete-time stochastic
quantum walk model focusing on a simple graph Z to see its fundamental property.
Comments 3: The possible physical implementations for the model could be discussed.
Response 3:Thank you for pointing this out. We agree with this comment. Therefore we inserted the following sentences in the last section on page 15 line 3 from the bottom (revised version).
We hope that in near future our observation can contribute the construction and analysis of a discrete-time version of quantum maze problem which is explored in continuous time stochastic quantum walk model in [20].
Comments 4: Some typos are found including equations print in line 167, line 132, etc.
Response 4:Thank you for pointing this out. We have improved then the following the referee's suggestion.
Reviewer 4 Report
Comments and Suggestions for Authors This is a neat piece of research where the authors propose a model to interpolate discrete-time quantum walks on the line and their persistent classical version. In particular, the ms shows that the limit distribution is described by the normal variance mixture with the arcsine law. I believe that this manuscript is worth of publication. My only reservation, given the diverse audience of Entropy, is that the authors did not provide concrete motivations for their analysis, not discuss potential application in quantum information.
Author Response
Comments 1: My only reservation, given the diverse audience of Entropy, is that the authors did not provide concrete motivations for their analysis, not discuss potential application in quantum information.
Response 1:Thank you for pointing this out. We agree with this comment. Therefore we inserted the following sentences in the last section on page 15 line 3 from the bottom (revised version).
We hope that in near future our observation can contribute the construction and analysis of a discrete-time version of quantum maze problem which is explored in continuous time stochastic quantum walk model in [20].